# A Novel Long Noncoding RNA in Osteocytes Regulates Bone Formation through the Wnt/β-Catenin Signaling Pathway

**DOI:** 10.3390/ijms241713633

**Published:** 2023-09-04

**Authors:** Makoto Arai, Hiroki Ochi, Satoko Sunamura, Nobuaki Ito, Masaomi Nangaku, Shu Takeda, Shingo Sato

**Affiliations:** 1Division of Nephrology and Endocrinology, The University of Tokyo Hospital, Tokyo 113-8655, Japan; 2Department of Rehabilitation for Motor Functions, Research Institute, National Rehabilitation Center for Persons with Disabilities, Tokorozawa 359-8555, Japan; 3Department of Orthopaedic Surgery, Graduate School, Tokyo Medical and Dental University (TMDU), Tokyo 113-8519, Japan; 4Osteoporosis Center, The University of Tokyo Hospital, Tokyo 113-8655, Japan; 5Division of Endocrinology, Toranomon Hospital Endocrine Center, Tokyo 105-8470, Japan; 6Center for Innovative Cancer Treatment, Tokyo Medical and Dental University (TMDU), Tokyo 113-8519, Japan

**Keywords:** long noncoding RNA, osteocyte, bone formation

## Abstract

The vast majority of transcribed RNAs are noncoding RNAs. Among noncoding RNAs, long noncoding RNAs (lncRNAs), which contain hundreds to thousands of bases, have received attention in many fields. The vast majority of the constituent cells in bone tissue are osteocytes, but their regulatory mechanisms are incompletely understood. Considering the wide range of potential contributions of lncRNAs to physiological processes and pathological conditions, we hypothesized that lncRNAs in osteocytes, which have not been reported, could be involved in bone metabolism. Here, we first isolated osteocytes from femurs of mice with osteocyte-specific GFP expression. Then, through RNA-sequencing, we identified osteocyte-specific lncRNAs and focused on a novel lncRNA, 9530026P05Rik (lncRNA953Rik), which strongly suppressed osteogenic differentiation. In the IDG-SW3 osteocyte line with lncRNA953Rik overexpression, the expression of *Osterix* and its downstream genes was reduced. RNA pull-down and subsequent LC-MS/MS analysis revealed that lncRNA953Rik bound the nuclear protein CCAR2. We demonstrated that CCAR2 promoted Wnt/β-catenin signaling and that lncRNA953Rik inhibited this pathway. lncRNA953Rik sequestered CCAR2 from HDAC1, leading to deacetylation of H3K27 in the *Osterix* promoter and consequent transcriptional downregulation of *Osterix*. This research is the first to clarify the role of a lncRNA in osteocytes. Our findings can pave the way for novel therapeutic options targeting lncRNAs in osteocytes to treat bone metabolic diseases such as osteoporosis.

## 1. Introduction

Human genome projects in the early 2000s revealed that only 2% of the human genome encodes proteins [1,2,3]. This finding raised the next question: what is the role of the other 98%? The subsequent ENCODE project demonstrated that more than 70% of the human genome is transcribed to RNAs but that most of these RNAs are noncoding RNAs [4]. These noncoding RNAs have been demonstrated to not be “junk” RNAs but to instead play important roles in biological phenomena. Among these RNAs, long noncoding RNAs (lncRNAs), containing hundreds to thousands of bases, are receiving increasing attention due to their various modes of action and their wide range of contributions to biological phenomena [5]. lncRNAs can modulate gene expression in multiple layers including epigenetic, transcriptional and translational regulations. lncRNAs can also interact with other kinds of RNAs such as mRNAs and miRNAs. To date, extensive research on lncRNAs has been made especially in the regions of cancer, development and metabolism [6,7,8].

Bone is recognized to be an organ where biological reactions occur. Formerly, bone was thought to be static, functioning only to support the body against gravity. However, bone has been shown to be dynamic, undergoing continuous remodeling and interactions with other organs. For example, osteocytes secrete FGF23 and modulate the serum phosphate concentration [9]. Osteocalcin, secreted from osteoblasts, acts on pancreatic beta cells to enhance insulin secretion [10], or on testicular Leydig cells to promote testosterone secretion [11]. These contributions are widely seen across species and bone plays important roles in systemic homeostasis.

Behind these advances in the understanding of bone metabolism lies the recent increase in the number of osteoporosis patients along with the aging of society. Osteoporosis increases the risk of fractures, which not only lead to the restriction of activities of daily living but also to a shortened lifespan [12,13,14,15]. To date, many studies on bone metabolism have been performed and have contributed to the understanding of the related biological phenomena and the development of anti-osteoporosis medications. Bone homeostasis is based on bone remodeling—a mechanism in which bone resorption by osteoclasts is coordinated with bone formation by osteoblasts. Osteoclasts and osteoblasts are key players in bone biology, but the vast majority of cells in bone tissue are not osteoclasts or osteoblasts but osteocytes. Osteocytes compose 90–95% of bone tissue, but their involvement in bone homeostasis has been underestimated. Osteocytes are derived from osteoblasts and become embedded beneath the bone matrix after differentiation. Recently, osteocytes have been reported to contact each other and other types of cells, such as osteoblasts, through their dendritic projections and to be involved in signal transduction. In addition to participating in direct contacts, osteocytes express or secrete specific kinds of proteins and modulate bone remodeling; for example, osteocyte expression of RANKL activates osteoclasts, and osteocyte secretion of sclerostin inhibits osteoblasts [16,17,18,19,20,21,22]. Osteocytes also sense gravity and play important roles in the maintenance of bone mass [23,24,25].

Regarding noncoding RNAs, some reports have shown that lncRNAs in osteoclasts or osteoblasts are involved in bone remodeling [26,27,28]. However, to date, there are no reports on lncRNAs in osteocytes. Considering the roles of osteocytes in bone biology described above, however, lncRNAs in osteocytes are also expected to be involved in bone homeostasis in a physiological or pathological manner. Thus, here, we aimed to identify lncRNAs in osteocytes and to elucidate their modes of action in bone remodeling.

## 2. Results

### 2.1. Identification and Characterization of lncRNAs in Osteocytes

To identify lncRNAs in osteocytes, we used *Dmp1*-Cre;*CAG-CAT-EGFP tg* mice, in which osteocytes express EGFP under the control of *Dmp1* promoter [16]. We first enzymatically digested femurs of *Dmp1*-Cre;*CAG-CAT-EGFP tg* mice after flushing the bone marrow. The cells were FACS-sorted into the EGFP-positive (osteocytes) and EGFP-negative fractions (Appendix A). We confirmed that the EGFP-positive fraction showed much higher expression of osteocyte marker genes, such as *Dmp1*, *Sost* and *Mepe*, than the EGFP-negative fraction (Appendix A). We extracted RNA from each fraction and analyzed the expression of lncRNAs through RNA-sequencing (RNA-seq). The differentially expressed lncRNAs in osteocytes are shown in Figure 1A. We calculated the fold change value as the ratio of the expression of each lncRNA in the EGFP-positive fraction to that in the EGFP-negative fraction. The heatmap in Figure 1A shows the lncRNAs with a fold change in expression of greater than 2.

Next, to investigate the functions of these lncRNAs in osteocytes, we amplified the sequences of some of the higher-ranked lncRNAs, as shown in the heatmap in Figure 1A. Then, we transduced each of these sequences into the murine osteocyte line, IDG-SW3, using the PiggyBac Transposon Vector System. After selection with puromycin, we established seven IDG-SW3 cell lines with an overexpression of each individual lncRNA representative of osteocytes. Mock cells transduced with the empty PiggyBac transposon vector were used as controls. We confirmed the stable overexpression of each lncRNA by qPCR (Figure 1B).

To focus on the effects of these seven lncRNAs on osteogenic differentiation, we treated the corresponding cells with osteogenic stimuli—ascorbic acid and β-glycerophosphate. Generally, these stimuli induce the expression of *Dmp1*, a marker gene of osteocytes, and in our IDG-SW3 cell lines, *Dmp1* expression can be detected as GFP expression because GFP is expressed under the control of the *Dmp1* promoter. These stimuli also promote mineralization, which is represented by Alizarin red S staining. Considering the results of the *Dmp1*-expression measurement, the results of Alizarin red S staining and the fold changes presented in the heatmap, we focused on a novel lncRNA, 9530026P05Rik (hereafter referred to as lncRNA953Rik). lncRNA953Rik had the second highest fold change among the lncRNAs represented in the heatmap (Figure 1A). IDG-SW3 cells overexpressing lncRNA953Rik showed weak expression of *Dmp1* and suppressed induction of mineralization under exposure to osteogenic stimuli (Figure 1C,D). Based on these results, the novel lncRNA lncRNA953Rik is highly specific to osteocytes, and osteogenic differentiation was suppressed in IDG-SW3 cells overexpressing lncRNA953Rik.

### 2.2. lncRNA953Rik Inhibits Osteogenesis

To investigate the modes of action of lncRNA953Rik in osteogenesis, we used IDG-SW3 cells overexpressing lncRNA953Rik and the corresponding control mock IDG-SW3 cells. In the alkaline phosphatase (ALP) activity assay, IDG-SW3 cells overexpressing lncRNA953Rik showed suppressed induction of ALP during osteogenic differentiation (Figure 2A). The qPCR, results showed that although the expression of *Runx2*, the farthest upstream regulator of osteogenesis, was comparable between control and lncRNA953Rik overexpressing cells, the expression of other downstream bone-formation markers, such as *Osterix*, *Alpl*, *Osteocalcin* and *Dmp1*, was lower in lncRNA953Rik-overexpressing cells (Figure 2B). Additionally, immunoblotting showed that the protein level of Osterix but not that of Runx2 was lower in lncRNA953Rik-overexpressing cells (Figure 2C). These results suggest that lncRNA953Rik overexpression suppresses osteogenesis through *Osterix* and its downstream targets.

In addition to examining the effect of lncRNA953Rik overexpression as described above, we examined the effect of lncRNA953Rik knockdown on differentiation. Using antisense oligonucleotides, we knocked down the endogenous expression of lncRNA953Rik in wild-type IDG-SW3 cells. Knocking down lncRNA953Rik did not change the expression of *Runx2* but increased the expression of *Osterix* and *Alpl*, as shown by qPCR and immunoblotting (Figure 2D,E). This result was supported by the finding of enhanced ALP activity and ALP staining in lncRNA953Rik-knockeddown IDG-SW3 cells (Figure 2F,G). Taken together, the results of these overexpression and knockdown experiments suggest that lncRNA953Rik inhibits osteogenesis.

### 2.3. lncRNA953Rik Binds to CCAR2

To explore how lncRNA953Rik inhibits osteogenesis, we first examined its localization. The results of fluorescence in situ hybridization (FISH) with probes for lncRNA953Rik showed that lncRNA953Rik was localized in the nucleus in IDG-SW3 cells (Figure 3A). Next, to comprehensively search for potential targets of lncRNA953Rik, we performed in vitro-transcription of lncRNA953Rik or a negative control RNA of the same length, and using these bait RNAs and whole-cell lysates from IDG-SW3 cells, we performed RNA pull-down followed by LC-MS/MS analysis (Appendix A). Shotgun proteomics analysis of the total eluted proteins identified 245 proteins that specifically bound to lncRNA953Rik (Figure 2B). 

Among these 245 proteins, we focused on nuclear proteins based on our FISH result and selected CCAR2, which was ranked second in the list. CCAR2 is a nuclear protein and is reported to be associated with the Wnt/β-catenin signaling pathway in cancer [29]. The Wnt/β-catenin signaling pathway is also one of the most important pathways in osteogenesis [30,31,32]. Therefore, we assumed that CCAR2 might be a reasonable candidate lncRNA953Rik-binding protein. We confirmed that lncRNA953Rik bound to CCAR2 by RNA pull-down followed by immunoblotting with an anti-CCAR2 antibody using either whole-cell lysates or nuclear extracts from IDG-SW3 cells, consistent with the results of the proteomics analysis (Figure 2C,D).

In addition, we reconfirmed this binding by the reverse approach. The results of RNA immunoprecipitation of nuclear extracts from IDG-SW3 cells with an anti-CCAR2 antibody or control IgG followed by qPCR with primers targeting lncRNA953Rik showed that CCAR2 bound to lncRNA953Rik (Figure 2E). These results of these bidirectional experiments demonstrated the binding of lncRNA953Rik to CCAR2 in nuclei.

### 2.4. lncRNA953Rik Suppresses Wnt/β-Catenin Signaling

We further sought to identify the signaling pathway through which lncRNA953Rik inhibits osteogenesis. Based on the results above showing the binding of lncRNA953Rik to CCAR2 and the reported association of CCAR2 with the Wnt/β-catenin signaling pathway, we hypothesized the association of lncRNA953Rik with the Wnt/β-catenin signaling pathway. In the Wnt/β-catenin signaling pathway, the binding of ligands such as Wnt3a to their receptors on cell membranes transduces signals into the cytoplasm, and these signals converge on the nuclear translocation of β-catenin and subsequent transcriptional upregulation of its target genes [33,34,35]. Our immunoblot analysis showed an increase in the level of non-phosphorylated β-catenin (non-phospho-β-catenin), namely, active β-catenin, after Wnt3a treatment in control IDG-SW3 cells; in contrast, in lncRNA953Rik-overexpressing IDG-SW3 cells, not only was the basal level of active β-catenin lower, but the increase in the level of active β-catenin after Wnt3a treatment was also attenuated (Figure 4A).

Consistent with this result, our immunoblot analysis using protein samples obtained after cytoplasmic and nuclear fractionations showed that non-phospho-β-catenin (i.e., active β-catenin) was localized in the nucleus and was less abundant in IDG-SW3 cells overexpressing lncRNA953Rik, while the level of phospho-β-catenin (i.e., inactive β-catenin) was comparable between control cells and lncRNA953Rik-overexpressing cells. Moreover, nuclear expression of Runx2 was comparable between the two groups, but nuclear expression of Osterix was lower in lncRNA953Rik-overexpressing IDG-SW3 cells (Figure 4B). These results were consistent with the results of immunoblot analysis of the whole-cell lysates and the qPCR analysis shown in Figure 2. We also investigated several representative genes known to suppress the Wnt/β-catenin signaling pathway. The expression levels of inhibitory genes, including *Dkk1*, *Krm1*, *Krm2*, *Sfrp1* and *Wif1*, were lower in IDG-SW3 cells overexpressing lncRNA953Rik (Appendix A), suggesting that the inhibitory effects of lncRNA953Rik on the Wnt/β-catenin signaling pathway are not due to upregulation of genes inhibiting the Wnt/β-catenin signaling but instead, the reduced expression of those genes is a compensatory response secondary to the suppression of the Wnt/β-catenin signaling pathway by lncRNA953Rik.

Next, we investigated the effect of CCAR2, whose function in osteogenesis is unknown. Overexpression of CCAR2 in IDG-SW3 cells increased ALP activity during osteogenic differentiation, while knockdown of CCAR2 by siRNA in IDG-SW3 cells suppressed ALP activity (Figure 4C,D). Knockdown of CCAR2 also suppressed the expression of Wnt/β-catenin signaling pathway marker genes, such as *Tcf7*, *Lef1* and *Axin2*, and several representative genes involved in bone formation, including *Osterix* and its downstream targets (Figure 4E,F).

Based on this stimulatory effect of CCAR2 on the Wnt/β-catenin signaling pathway, we investigated the interaction of CCAR2 and lncRNA953Rik using reporter assays to monitor Wnt/β-catenin signaling pathway activity with TOP-Flash and FOP-Flash plasmids. Wnt3a treatment increased reporter activity in control IDG-SW3 cells, but the increase was suppressed in IDG-SW3 cells overexpressing lncRNA953Rik (Figure 4G). LiCl is also a stimulant of the Wnt/β-catenin signaling pathway. It inhibits GSK-3β in the cytoplasm, leading to the inhibition of β-catenin degradation [36,37]. As observed with Wnt3a treatment, the increase in the reporter activity induced by LiCl was suppressed in IDG-SW3 cells overexpressing lncRNA953Rik (Figure 4H). Regarding CCAR2, the overexpression of CCAR2 in IDG-SW3 cells enhanced LiCl-induced activation of the Wnt/β-catenin signaling pathway (Figure 4I). Based on these results, we co-transfected lncRNA953Rik and CCAR2 into IDG-SW3 cells. The increase in the reporter activity induced by LiCl was suppressed by lncRNA953Rik, but this suppressive effect was reversed by co-transfection of CCAR2 (Figure 4J). These results suggest that lncRNA953Rik and CCAR2 inhibit and promote the Wnt/β-catenin signaling pathway, respectively, through the same pathway, resulting in the regulation of the expression of *Osterix* and its downstream targets.

### 2.5. lncRNA953Rik Suppresses Osterix Transcription by Sequestering CCAR2 from HDAC1

The results thus far showed that lncRNA953Rik and CCAR2 modulate osteogenesis through the Wnt/β-catenin signaling pathway. Then, we investigated their molecular mechanisms in more detail. A previous report showed that CCAR2 binds to histone deacetylase 1 (HDAC1) and HDAC3 in HEK293T cells [38]. HDACs are involved in the Wnt/β-catenin signaling pathway; thus, we hypothesized that HDACs are also important modulators in the currently investigated context. Immunoprecipitation of nuclear extracts from IDG-SW3 cells with an anti-CCAR2 antibody showed that CCAR2 bound to HDAC1 and HDAC2 but not HDAC3 (Figure 5A). Reciprocally, immunoprecipitation with an anti-HDAC1 or anti-HDAC2 antibody showed that HDAC1 but not HDAC2 bound to CCAR2 (Figure 5B,C).

Based on these results, we performed further analyses focusing on CCAR2 and HDAC1 to clarify how lncRNA953Rik affects the binding of CCAR2 and HDAC1. Immunoprecipitation of nuclear extracts from control or lncRNA953Rik-overexpressing IDG-SW3 cells with an anti-CCAR2 antibody showed that the binding of CCAR2 and HDAC1 was weaker in lncRNA953Rik-overexpressing IDG-SW3 cells, although the total amounts of nuclear CCAR2 and HDAC1 did not differ between the cell lines (Figure 5D).

Then, we investigated how the binding of CCAR2 and HDAC1 is involved in the transcriptional regulation of *Osterix*. The promoter region of *Osterix* contains two binding sites for TCF/LEF, activators of the Wnt/β-catenin signaling pathway: CTTTGGG sequences located 2022 and 222 bases upstream of the transcription start site [39] (Figure 5E). Chromatin immunoprecipitation followed by qPCR (ChIP-qPCR) using primers targeting the TCF/LEF-binding sites in the *Osterix* promoter region showed that the acetylation level of histone 3 lysine 27 (H3K27), a marker of active promoter regions, was lower in IDG-SW3 cells overexpressing lncRNA953Rik, although the recruitment of HDAC1 was comparable between control and lncRNA953Rik-overexpressing IDG-SW3 cells (Figure 5F,G). The recruitment of CCAR2 to these regions was reduced in IDG-SW3 cells overexpressing lncRNA953Rik (Figure 5H). Considering our results showing that CCAR2 activates the Wnt/β-catenin signaling pathway and binds to HDAC1, these ChIP-qPCR results suggest that lncRNA953Rik binds to CCAR2 and sequesters it from HDAC1, resulting in uninhibited HDAC1 activity, in turn leading to deacetylation of H3K27 in the *Osterix* promoter region and the consequent transcriptional downregulation of *Osterix* (Figure 6).

## 3. Discussion

Osteocytes are the most abundant cells in bone tissue, and evidence of their importance in multiple functions has recently been accumulating. The involvement of lncRNAs in multiple organs has also been revealed, but their functions in osteocytes remain unknown. In this research, we aimed to identify lncRNAs regulating osteocyte functions. Through comprehensive searches in osteocytes, we identified several novel lncRNAs enriched in osteocytes. Based on the effects of these lncRNAs on osteoblasts differentiation to osteocytes, a novel lncRNA, lncRNA953Rik, whose function was unknown in any tissue or cell, including osteocytes, was found to strongly inhibit osteogenic differentiation. Further mechanistic explorations revealed that lncRNA953Rik inhibited the transcription of *Osterix* through epigenetic regulation, resulting in the suppression of osteoblast differentiation to osteocytes.

To isolate osteocytes, we performed a series of enzymatic digestions followed by FACS sorting of EGFP-positive cells from the femurs of *Dmp1*-Cre;*CAG-CAT-EGFP tg* mice. This fluorescence-based sorting strategy enabled us to selectively collect osteocytes, which are hard to isolate by only surface marker expression. Through comprehensive searches using RNA-seq data, we identified representative but unknown lncRNAs in osteocytes. 

Then, we focused on bone formation because there are large unmet needs in therapeutic options for osteoporosis that target bone formation. Anti-osteoporotic medications have redressed the imbalance between bone resorption and bone formation, and have proved effective, but most of these agents are anti-resorptive agents. There are only two options for targeting bone formation: parathyroid hormone (PTH) analogs and anti-sclerostin antibodies. In addition, both of these options can be used only for limited periods due to their potential side effects [40,41,42]. Considering this medical context, we aimed to identify lncRNAs in osteocytes that regulate bone formation, and we finally identified a novel lncRNA, lncRNA953Rik, which ranked high in the RNA-seq data and strongly inhibited bone formation.

When we investigated the targets of lncRNA953Rik, we found that the expression of *Runx2*, a master regulator of bone formation, was not affected by lncRNA953Rik, but the expression of *Osterix*, another important regulator of bone formation, and its downstream genes was reduced by lncRNA953Rik. This decrease in expression was observed at both the transcriptional and translational levels. *Osterix* has been considered to be downstream of *Runx2*, but recent reports have shown that *Osterix* activates the enhancers of *Runx2* [43]; moreover, some genes are *Runx2*-dependent and *Osterix*-independent, and other genes are *Runx2*-independent and *Osterix*-dependent [44]. Additionally, both *Runx2* and *Osterix* are target genes in the Wnt/β-catenin signaling pathway, and some reports have shown that *Osterix* can function as a direct target of the Wnt/β-catenin signaling pathway, not through *Runx2* [45,46]. These reports suggest that the functions and interactions of *Runx2* and *Osterix* in bone formation are complex, and they remain to be elucidated in both physiological and pathological states.

In addition to *Runx2*, there are many mediators reported to activate *Osterix* such as the mitogen-activated protein kinase (MAPK) pathway, extracellular signal-regulated kinase (Erk) and miRNAs [47]. In our current study, we focused on Wnt/β-catenin signaling pathway based on our comprehensive proteomics analysis and previous reports, but we need to note that there can be other potential genes and pathways which regulate *Osterix* in the mode of action of lncRNA953Rik.

One of the greatest challenges in the research on lncRNAs is identifying the mechanisms of lncRNAs. lncRNAs can function in several ways [5]. Here, we showed that lncRNA953Rik bound to the nuclear protein CCAR2 and modulated its functions. In addition to investigating its interaction with proteins, we also investigated other possible mechanisms of action of lncRNA953Rik. The first possible mechanism is transcriptional regulation, which often occurs in *cis*, i.e., on the same chromosome. Here, we generated IDG-SW3 cells overexpressing lncRNA953Rik using a transposon, which results in the insertion of genes into chromosomes at random locations. Therefore, it is likely that lncRNA953Rik was integrated by the transposon into different chromosomes than those hosting endogenous lncRNA953Rik and thus acted in *trans*. Furthermore, the IDG-SW3 cells overexpressing lncRNA953Rik were generated through bulk culture, not from single colonies; thus, it is unlikely that lncRNA953Rik was consistently integrated into the same chromosome hosting endogenous lncRNA953Rik. The second possible mechanism is functioning as an antisense transcript. *Adamts9* is located on the opposite strand with respect to lncRNA953Rik, but the expression of *Adamts9* was not related to that of lncRNA953Rik. The third possible mechanism is the cleavage of small RNAs from within lncRNAs. Small RNAs such as miRNAs are reported to be generated from lncRNAs by internal cleavage of the lncRNAs, but no small RNA sequences in the sequence of lncRNA953Rik have been reported. Another possible mechanism is the encoding of proteins by lncRNAs. Although lncRNAs are “noncoding” RNAs by definition, some lncRNAs encode proteins [48]. Here, we amplified several potential protein-coding sequences beginning at a start codon and ending at a stop codon within lncRNA953Rik and fused these sequences to a FLAG tag at the 3’ end. We subsequently confirmed that no proteins were expressed from these constructs. Based on these results, we concluded that lncRNA953Rik acts through binding to proteins and subsequently modulating their functions (Figure 3). 

Our comprehensive searches for the proteins binding to lncRNA953Rik using RNA pull-down followed by mass spectrometry revealed several candidates. Among those proteins, we focused on the nuclear protein CCAR2 due to its higher rank in the list of candidates, its association with the Wnt/β-catenin signaling pathway [29] and our FISH results showing that lncRNA953Rik was located in the nucleus. The Wnt/β-catenin signaling pathway is essential for skeletal development and is particularly important for bone formation. Although the importance of CCAR2 in the Wnt/β-catenin signaling pathway was unknown, our results showed that CCAR2 promoted signaling through this pathway and facilitated bone formation and that lncRNA953Rik bound to CCAR2 and inhibited its stimulatory effects, leading to an epigenetic modulation of histone acetylation involved in the transcriptional regulation of *Osterix*. Although further studies are necessary to reveal the details of the mechanism incorporating CCAR2 and HDAC1, our research is the first to identify representative lncRNAs in osteocytes and reveal their epigenetic regulation of bone formation.

Upstream factors that up- or downregulate the expression of lncRNA953Rik remain unknown. Many medications, hormones such as PTH, or pathological conditions such as menopause or aging, can modulate bone homeostasis, and further studies are necessary to identify the factors modulating the expression of lncRNA953Rik. However, our research identified lncRNA953Rik as representative of osteocytes and elucidated its epigenetic mode of action in detail. Based on our results showing the effectiveness of antisense oligonucleotides targeting lncRNA953Rik, lncRNA953Rik can be exploited as a therapeutic target in the Wnt/β-catenin signaling pathway and bone formation. Similarly, our findings can pave the way for the development of novel therapeutic options targeting lncRNAs in osteocytes for bone metabolic diseases such as osteoporosis.

## 4. Materials and Methods

### 4.1. Cells and Reagents

IDG-SW3 cells, murine osteocyte cell line, were kindly given by Prof. Lynda F Bonewald at Indiana Center for Musculoskeletal Health, Indiana University School of Medicine, Indianapolis, IN, USA. Cells were cultured in MEMα medium containing nucleotides (Thermo Fisher Scientific, Waltham, MA, USA), 10% FBS, 1% penicillin and streptomycin and 50 U/mL IFN-γ (Thermo Fisher Scientific) on type-I-collagen-coated plastic dishes at 33 degrees [49]. Media were changed every 2–3 days. Osteogenic differentiation was induced by 50 µg/mL ascorbic acid (MilliporeSigma, Burlington, MA, USA) and 4 mM β-glycerophosphate (nacalai, Kyoto, Japan).

To stimulate the Wnt/β-catenin signaling pathway, recombinant mouse Wnt3a (Bio-Techne, Minneapolis, MN, USA) was added to media at a final concentration of 100 ng/mL.

To generate IDG-SW3 cells constitutively expressing a lncRNA, we amplified the lncRNA and inserted it into Super PiggyBac Transposase Expression Vector (System Biosciences, Palo Alto, CA, USA, #PB210PA-1) and transduced it into IDG-SW3 cells by electroporation. Electroporation was performed by Neon (Thermo Fisher Scientific) according to the manufacturer’s instructions with 2.4 µg Super PiggyBac Transposase Expression Vector, with each lncRNA and 0.6 µg transposase per 1.5 × 10^5^ IDG-SW3 cells. After electroporation, cells were cultured in bulk without single-cell cloning and passaged before reaching confluency. Puromycin was added at every other passage at a final concentration of 2 µg/mL. After culturing for 3 weeks, we obtained IDG-SW3 cells constitutively expressing each lncRNA. Empty Super PiggyBac Transposase Expression Vector without lncRNAs was used to generate mock cells. The overexpression of each lncRNA was confirmed by qPCR.

For transient overexpression, we used a reagent for transfection, TransIT-X2 (Mirus Bio, Madison, WI, USA).

### 4.2. Alkaline Phosphatase (ALP) Assay

Cells cultured in 24-well plates were rinsed by 50 mM Tris-HCl (pH 7.4) and dissolved by 0.2% Triton X-100. After centrifugation, supernatant was used for the assay. After incubated with 3.72 mg/mL 4-nitrophenyl phosphate disodium salt hexahydrate (Fujifilm, Osaka, Japan), 1 mM MgCl_2_ and 56 mM 2-amino-2-methyl-1,3-propanediol (pH 10.0) at 37 degrees for 30 min, the absorbance at 405 nm was measured. The concentration of the protein of the same sample was measured using the bicinchoninic acid method. Finally, we calculated ALP activity (nmol p-nitrophenol/min/mg protein) by dividing the absorbance at 405 nm by the protein concentration [50].

### 4.3. ALP Staining

Cells cultured in 12-well plates were fixed by 4% paraformaldehyde and treated by 0.1 mg/mL naphthol AS-MX phosphate (MilliporeSigma), 2 mM MgCl_2_, 0.1 M Tris-HCl (pH 8.8), N,N-dimethylformamide and 0.6 mg/mL Fast red violet LB salt (MilliporeSigma) at 37 degrees for 20 min. The images were taken after several washes.

### 4.4. Alizarin Red S Staining

Cells cultured in 12-well plates were fixed by 4% paraformaldehyde and treated by 1% Alizarin red S solution (MilliporeSigma) for 30 min in a dark place. The images were taken after several washes [51].

### 4.5. RNA Extraction, Reverse Transcription and qPCR

Cells harvested in TRIzol (Thermo Fisher Scientific) were subjected to RNA extraction procedures according to the manufacturer’s instructions. Briefly, each sample in 1 mL TRIzol was mixed with 0.2 mL chloroform. After centrifugation, the aqueous layer was collected and mixed with the same amount of 100% ethanol and then RNA was extracted with NucleoSpin RNA column methods (Takara, Kusatsu, Japan). This method included the treatment of DNase I.

Based on the RNA concentrations measured by NanoDrop (Thermo Fisher Scientific), each RNA sample was used for reverse transcription reactions with ReverTra Ace qPCR RT Master Mix with gDNA Remover (Toyobo, Osaka, Japan) according to the manufacturer’s instructions. qPCR was performed with Brilliant III Ultra-Fast QPCR Master Mix (Agilent Technologies, Santa Clara, CA, USA). The protocol was 15 min at 95 degrees, 40 cycles of 10 s at 95 degrees and 30 s at 60 degrees, followed by a dissociation stage (1 min at 95 degrees, 30 s at 55 degrees and 30 s at 95 degrees). The results after normalization to *Gapdh* are shown in the figures. The sequences of the primers are as follows (Table 1).

### 4.6. Immunoblotting

Cells were harvested in TNE buffer (10 mM Tris-HCl, pH 7.8, 150 mM NaCl, 1 mM EDTA, 1% NP-40) with cOmplete protease inhibitor cocktail (Roche, Basel, Switzerland). After centrifugation, the supernatant was collected as whole-cell lysates and used for immunoblotting. Based on the protein concentrations measured through the bicinchoninic acid method, each sample was boiled with Sample Buffer Solution with Reducing Reagent (6×) for SDS-PAGE (nacalai) and run in the gel for SDS-PAGE. After electrophoresis, the gel was transferred to the PVDF membrane. The membrane was immersed in 5% skim milk and incubated with the first antibody at 4 degrees overnight. The next day, the membrane was incubated with the second antibody and the bands were detected with Chemi-Lumi One Ultra (nacalai). Relative protein levels were measured by densitometry.

The first antibodies used are as follows. Acetyl-Histone H3 (MilliporeSigma, #06-599), ALPL (GeneTex, Irvine, CA, USA, GTX62596), CCAR2 (Medical & Biological Laboratories, Tokyo, Japan, RN117PW), GAPDH (Cell Signaling Technology, Danvers, MA, USA, #2118), HDAC1 (Cell Signaling Technology, #5356), HDAC1 (Cell Signaling Technology, #34589), HDAC2 (Cell Signaling Technology, #5113), HDAC3 (Cell Signaling Technology, #3949), Histone H3 (Abcam, Cambridge, UK, ab1791), Non-phospho-β-catenin (Ser33/37/Thr41) (Cell Signaling Technology, #4270), Osterix (Abcam, ab22552), Phospho-β-catenin (Ser33/37/Thr41) (Cell Signaling Technology, #9561) and Runx2 (Cell Signaling Technology, #8486).

### 4.7. Cell Fractionation

Cells were harvested in hypotonic buffer (10 mM HEPES-NaOH, pH 7.5, 1.5 mM MgCl_2_, 10 mM KCl, 1 mM EDTA, 1 mM EGTA) with cOmplete protease inhibitor cocktail (Roche) and 1 mM PMSF. Cells were resuspended through 23-gauge needles. After centrifugation, the supernatant was further ultracentrifuged to obtain cytoplasmic fractions. The precipitate was resuspended with SDS lysis buffer (50 mM Tris-HCl, pH 8.0, 10 mM EDTA, 1% SDS). After centrifugation, the supernatant was collected as nuclear extracts.

### 4.8. Antisense Oligonucleotide-Mediated Knockdown

Antisense oligonucleotides targeting lncRNA953Rik and its negative control (Antisense LNA GapmeR; Exiqon, Woburn, MA, USA) were transfected into IDG-SW3 cells at a final concentration of 50 nM through TransIT-X2 (Mirus). The sequences of the oligonucleotides are as follows (Table 2).

### 4.9. siRNA-Mediated Knockdown

siRNAs targeting *Ccar2* (Thermo Fisher Scientific) and its negative control (BLOCK-iT Alexa Fluor Red, Thermo Fisher Scientific) were transfected into IDG-SW3 cells at a final concentration of 50 nM through TransIT-X2 (Mirus). The sequences of the siRNAs are as follows (Table 3). 

### 4.10. Mice

We used *Dmp1*-Cre;*CAG-CAT-EGFP tg* mice, in which osteocytes express EGFP under the control of the *Dmp1* promoter. The mice were generated by mating *CAG-CAT-EGFP tg* mice [52] and *Dmp1*-Cre mice (derived from Prof. Lynda F Bonewald at Indiana University IN, USA) [53]. 

### 4.11. Isolation of Osteocytes

We collected the femurs of *Dmp1*-Cre;*CAG-CAT-EGFP tg* mice, cut both ends of the femurs and removed the bone marrow inside. The femurs were incubated in digestion medium 1 (2 mg/mL type 2 collagenase, 0.05% trypsin in MEMα medium) with shaking at 200 rpm at 37 degrees for 30 min. After the supernatant was removed, the femurs were again incubated in digestion medium 1 with shaking at 200 rpm at 37 degrees for 60 min. The supernatant was harvested and after centrifugation, the precipitate was resuspended with culture medium (MEMα, 10% FBS and 1% penicillin and streptomycin) and stored at 4 degrees as Fraction 1.

The femurs were then cut into pieces by scissors in MEMα medium on a Petri dish. MEMα medium was harvested and centrifuged. The precipitate was resuspended with culture medium and stored at 4 degrees as Fraction 2.

The bone tissue was further incubated in digestion medium 1 with shaking at 200 rpm at 37 degrees for 30 min. The supernatant and PBS used for washing were centrifuged and the precipitate was resuspended with culture medium and stored at 4 degrees as Fraction 3.

The bone tissue was then incubated in digestion medium 2 (5 mM EDTA and 0.05% trypsin in MEMα medium) with shaking at 200 rpm at 37 degrees for 30 min. The supernatant and PBS used for washing were centrifuged and the precipitate was resuspended with culture medium and stored at 4 degrees as Fraction 4.

In the same way as Fraction 3, Fraction 5 was obtained.

Fractions 1 to 5 were mixed through a cell strainer and centrifuged. The precipitate was resuspended with PBS and used for FACS sorting. EGFP-positive or negative cells were collected and the total RNA was extracted from each cellular fraction.

### 4.12. RNA-seq

The isolation of osteocytes was performed twice as described in 4.11. From each 1 µg RNA from four RNA samples (two from the EGFP-positive and two from the EGFP-negative cells), the libraries for RNA-seq were constructed. The sequencing was performed through MiSeq (Illumina, San Diego, CA, USA). The sequence data were mapped to the reference mouse genome. Annotation was carried out with reference to the Ensembl database. The expression of each lncRNA registered in the database was calculated in terms of fragments per kilobase of exon per million reads mapped (FPKM).

### 4.13. FISH

FISH experiments were performed according to the protocol of Stellaris RNA FISH (Biosearch Technologies, Hoddesdon, UK). Briefly, IDG-SW3 cells were fixed by 3.7% formaldehyde and permeabilized by 70% ethanol at 4 degrees for one hour. After washing, the cells were treated with probe solutions (125 nM probes and 10% formaldehyde) at 37 degrees for 16 h in the dark. After washing, the cells were stained by DAPI and observed with confocal microscopy (Nikon, Tokyo, Japan). The probes for lncRNA953Rik were synthesized according to the Stellaris RNA FISH probe designer.

### 4.14. RNA Pull-Down

From a pcDNA3-lncRNA953Rik plasmid or an original pcDNA3 plasmid, we PCR-amplified the double-stranded linear DNAs with a T7 promoter sequence at the 5′ end of lncRNA953Rik or an irrelevant DNA sequence with the same number of bases as lncRNA953Rik (negative control), respectively. The sequences of the primers are as follows (Table 4). 

From these double-stranded linear DNAs, lncRNA953Rik or negative control RNAs were in vitro-transcribed at 37 degrees for two hours (in vitro transcription T7 Kit; Takara). During this in vitro transcription, Biotin-16-UTPs (Roche) were incorporated at a ratio of Biotin-16-UTP to UTP as 1 to 3. 

With these biotin-labeled RNAs as baits, RNA pull-down was performed according to the MagCapture RNA Pull-Down Assay Kit protocol (Fujifilm). Briefly, after being denatured at 90 degrees for two minutes, biotin-labeled RNAs were mixed with IDG-SW3 cell lysates and rotated at 4 degrees for three hours. Then, magnetic beads (MagCapture HP Tamavidin 2-REV; Fujifilm) were added and rotated at 4 degrees for one hour. After several washes, RNA-binding proteins were eluted under a nondenaturing condition through the reactions with excessive amounts of biotin solutions at room temperature for ten minutes.

### 4.15. Silver Staining

After SDS-PAGE, the gels were fixed by 25% ethanol and 5% acetic acid for two hours, sensitized by 0.02% sodium thiosulfate for one minute and reacted with 0.1% silver nitrate for 30 min. The gels were developed by 0.036% formaldehyde and 2% sodium carbonate and the reaction was stopped by 2% acetic acid.

### 4.16. Mass Spectrometry

The eluates after RNA pull-down were analyzed through LC-MS/MS shotgun proteomics by APRO Science (Tokushima, Japan). The samples were incubated with trichloroacetic acid and the precipitates were dissolved by Tris buffer (250 mM Tris-HCl, pH 8.5, 2 mM EDTA). Based on the protein concentrations measured through the bicinchoninic acid method, 2 µg protein was reduced by 2 µL of 0.67 M dithiothreitol (DTT) in Tris buffer at 37 degrees for two hours. The products were alkylated by 2 µL of 1.4 M iodoacetamide in Tris buffer at room temperature for 30 min under light-shielding conditions. The products were digested by trypsin in Tris buffer (pH 8.0) at 37 degrees for two hours. The products were desalinated and analyzed by LC-MS/MS. Ionization was performed by Nanoflow-LC ESI and analyzed by Q Exactive Plus (Thermo Fisher Scientific). Product ion spectrum data were reviewed in Mascot Server with the SwissProt mouse database and the information on identified proteins was saved in the Scaffold format.

### 4.17. RNA Immunoprecipitation (RIP)

Harvested cells were resuspended with Nuclear isolation buffer (1.28 M sucrose, 40 mM Tris-HCl, pH 7.5, 20 mM MgCl_2_, 4% Triton X-100) and incubated on ice for 20 min. After centrifugation, the precipitate was resuspended with RIP buffer (150 mM KCl, 25 mM Tris-HCl, pH 7.5, 5 mM EDTA, 0.5 mM DTT, 0.5% NP-40) with 0.4 U/µL RNase inhibitor (Takara) and cOmplete protease inhibitor cocktail (Roche) through 23-gauge needles. After centrifugation, the supernatant was collected as a nuclear fraction and rotated at 4 degrees overnight with Dynabeads Protein G (Thermo Fisher Scientific) which was incubated with antibodies at room temperature for one hour beforehand [54]. The next day the beads were washed with RIP buffer and PBS. Then, the beads were incubated with 0.05 U/µL DNase I (Takara) at 37 degrees for 20 min. TRIzol LS was added and RNA was extracted according to the manufacturer instructions. Extracted RNA was reverse transcribed to cDNA and used for qPCR. The details of reverse transcription, qPCR and primers were the same as described in 4.5. The antibodies used are as follows: CCAR2 (GeneTex, GTX32552) and Normal Rabbit IgG (Cell Signaling Technology, #2729).

### 4.18. Luciferase Assay

Cells were seeded on 24-well plates at 4.0 × 10^4^ cells/well. The next day, 300 ng TOP-Flash (TCF Reporter Plasmid) or FOP-Flash (TOP flash mutant) were transfected. At the same time, 0.5 ng pGL4.74 hRluc/TK Renilla luciferase plasmid was transfected into all the cells. An amount of 100 ng/mL Wnt3a or 20 mM LiCl was also added to culture medium to stimulate the Wnt/β-catenin signaling pathway. The luminescence was analyzed with a Dual-Luciferase Reporter Assay System (Promega, Madison, WI, USA). The values of luminescence of TOP-Flash or FOP-Flash were normalized to those of Renilla, and TOP/FOP ratios were calculated by dividing the normalized TOP-Flash values by the normalized FOP-Flash values.

### 4.19. Immunoprecipitation

Harvested cells were resuspended with Hypotonic buffer (10 mM HEPES-NaOH, pH 7.5, 1.5 mM MgCl_2_, 10 mM KCl) through 25-gauge needles. After centrifugation, the precipitate was resuspended with buffer containing 50 mM Tris-HCl, pH 8.0, 0.1 mM EDTA, 5% Glycerol, 100 mM KCl and 0.1% NP-40. After centrifugation, the supernatant was collected as a nuclear fraction and rotated at 4 degrees overnight with Dynabeads Protein G (Thermo Fisher Scientific) which was incubated with antibodies at room temperature for 30 min beforehand. The next day, after several washes, the proteins were eluted by SDS lysis buffer and used for immunoblotting. The antibodies used are as follows. CCAR2 (Medical & Biological Laboratories, RN117PW), HDAC1 (Cell Signaling Technology, #34589), HDAC2 (Cell Signaling Technology, #57156), Normal Rabbit IgG (Cell Signaling Technology, #2729) and Rabbit mAb IgG Isotype Control (Cell Signaling Technology, #3900).

### 4.20. Chromatin Immunoprecipitation (ChIP)

Cells were fixed by 1.5 mM ethylene glycol bis (succinimidyl succinate) (Thermo Fisher Scientific) at room temperature for 30 min followed by 1% formaldehyde for 10 min. The fixation was stopped by glycine and the cells were harvested. Harvested cells were resuspended with Hypotonic buffer (10 mM HEPES-NaOH, pH 7.5, 1.5 mM MgCl_2_, 10 mM KCl, 1 mM EDTA, 1 mM EGTA) through 23-gauge needles. After centrifugation, the precipitate was resuspended in Buffer A (0.32 mM sucrose, 15 mM HEPES-NaOH, pH 7.5, 60 mM KCl, 2 mM EDTA, 0.5 mM EGTA) and then Buffer B (877 mM sucrose, 15 mM HEPES-NaOH, pH 7.5, 60 mM KCl, 2 mM EDTA, 0.5 mM EGTA) was layered over Buffer A. After centrifugation, the precipitate was resuspended with buffer containing 15 mM HEPES-NaOH, pH 7.5, 60 mM KCl, 15 mM NaCl, 0.32 mM sucrose and 4 mM MgCl_2_, and incubated with Micrococcal Nuclease (Cell Signaling Technology) and 2 mM CaCl_2_ at 37 degrees for 20 min. The digestion was stopped by EDTA. After centrifugation, the precipitate was resuspended with buffer containing 20 mM Tris-HCl, pH 8.0, 2 mM EDTA, 1% Triton X-100, 150 mM NaCl and 0.1% SDS. After sonication and centrifugation, the supernatant was collected as input. The input samples were rotated at 4 degrees overnight with Dynabeads Protein G (Thermo Fisher Scientific) which was incubated with antibodies at room temperature for one hour beforehand. The next day, the beads were washed sequentially by the buffers: (1) 150 mM NaCl and 20 mM Tris-HCl, pH 8.0; (2) 500 mM NaCl, 20 mM Tris-HCl, pH 8.0, 2 mM EDTA, 1% Triton X-100 and 0.1% SDS; (3) 250 mM LiCl, 10 mM Tris-HCl, pH 8.0, 1 mM EDTA, 1% NP-40 and 1% sodium deoxycholate; and (4) TE buffer. Then, the beads were incubated with 1 mg/mL pronase (Fujifilm) at 65 degrees overnight. The next day, DNA was collected through Wizard SV Gel and PCR Clean-Up System (Promega) and used for ChIP-qPCR.

The antibodies used for ChIP are as follows: CCAR2 (GeneTex, GTX32552), H3K27Ac (Cell Signaling Technology, #8173), HDAC1 (Cell Signaling Technology, #34589), Normal Rabbit IgG (Cell Signaling Technology, #2729) and Rabbit mAb IgG Isotype Control (Cell Signaling Technology, #3900). 

The sequences of the primers for ChIP-qPCR are as follows (Table 5).

### 4.21. Statistics

Data are expressed as mean with error bars representing standard deviation. Statistical analyses were performed using a *t* test when comparing two groups and by ANOVA followed by Dunnett’s test or the SNK test when comparing more than two groups. *p* < 0.05 was considered statistically significant.

## Figures and Tables

**Figure 1 ijms-24-13633-f001:**
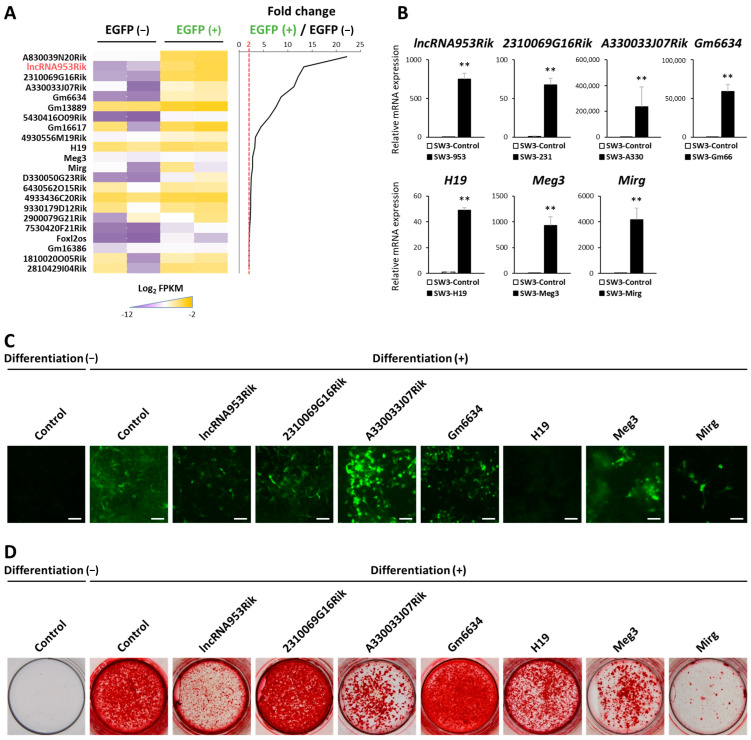
Identification of lncRNAs in osteocytes. (**A**) Femurs of *Dmp1*-Cre;*CAG-CAT-EGFP tg* mice were enzymatically digested after flushing the bone marrow. Cells were FACS-sorted into the EGFP-positive (osteocytes) and EGFP-negative fractions. These fractions were analyzed by RNA-seq and the heatmap shows the differentially expressed lncRNAs in osteocytes. *n* = 2. (**B**) Representative lncRNAs were transduced into IDG-SW3 cells. Stable overexpression of each lncRNA was confirmed by qPCR. The indicated RNA expression levels were normalized to the level of *Gapdh*. Error bars, SDs; *n* = 4; *t* test; **, *p* < 0.01 versus control (mock). (**C**,**D**) IDG-SW3 cells overexpressing each lncRNA were treated with differentiation medium for 28 days. Green fluorescence indicates the expression of GFP under the control of the *Dmp1* promoter. Bars, 100 µm. (**C**). Alizarin red S staining shows mineralized nodules (**D**).

**Figure 2 ijms-24-13633-f002:**
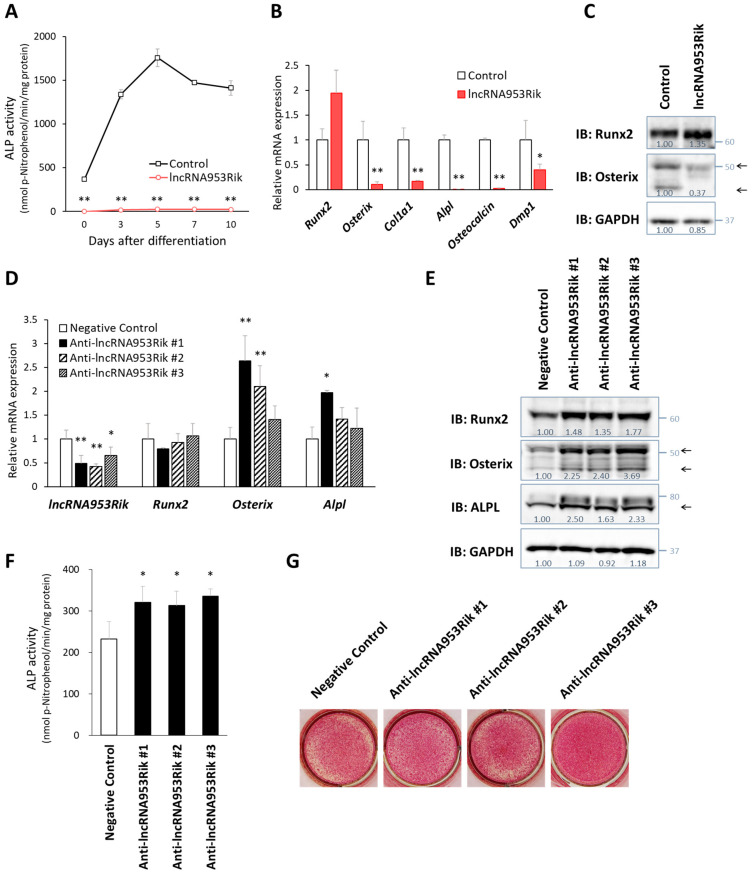
lncRNA953Rik inhibits osteogenesis. (**A**) ALP activity in lncRNA953Rik-overexpressing or mock IDG-SW3 cells (control) was measured over 10 days of culture with differentiation medium. Error bars, SDs; *n* = 3; *t* test; **, *p* < 0.01 versus control. (**B**,**C**) The expression of marker genes of bone formation was analyzed by qPCR (**B**) and immunoblotting (**C**) in the absence of differentiation. The indicated RNA expression levels were normalized to the level of *Gapdh*. Error bars, SDs; *n* = 3; *t* test; *, *p* < 0.05; **, *p* < 0.01 versus control (**B**). The arrows show the locations of Osterix (**C**). (**D**–**G**) Differentiation was induced in IDG-SW3 cells with differentiation medium on day 0. An antisense oligonucleotide targeting lncRNA953Rik (#1, #2 or #3) or a negative control antisense oligonucleotide was transduced on days 0, 3 and 6. Samples were obtained on day 9 and subjected to qPCR analysis (**D**), immunoblot analysis (**E**), an ALP activity assay (**F**) and ALP staining (**G**). The indicated RNA expression levels were normalized to the level of *Gapdh*. Error bars, SDs; *n* = 3; ANOVA followed by Dunnett’s test; *, *p* < 0.05; **, *p* < 0.01 versus control (**D**). The arrows show the locations of Osterix and ALPL (**E**). Error bars, SDs; *n* = 3; ANOVA followed by Dunnett’s test; *, *p* < 0.05 versus control (**F**). Values under each group of immunoblot analysis are relative protein levels measured by densitometry.

**Figure 3 ijms-24-13633-f003:**
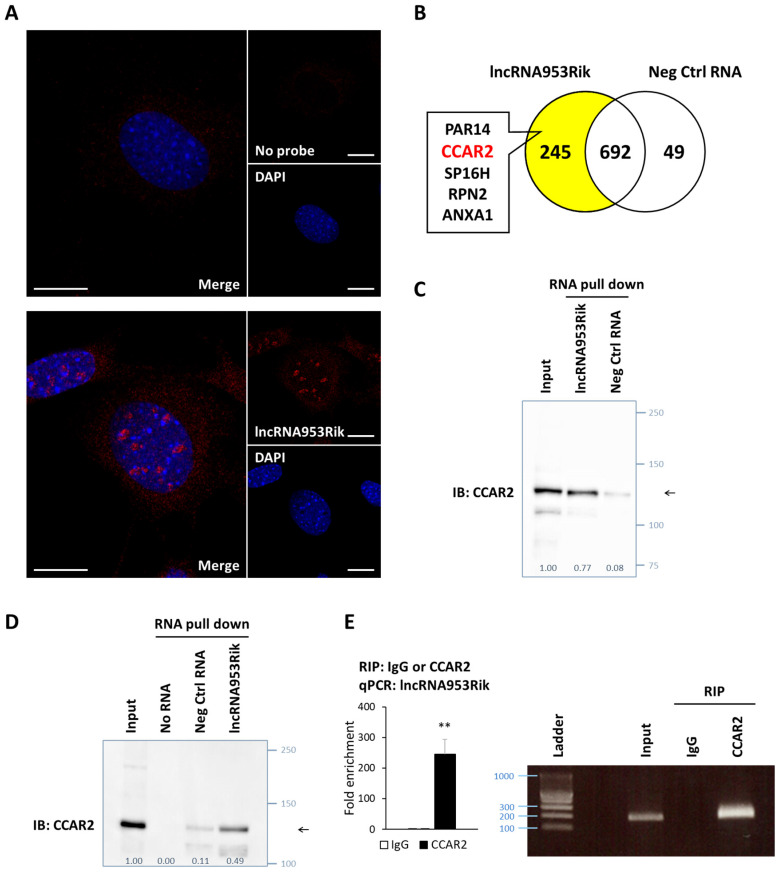
lncRNA953Rik binds to CCAR2. (**A**) IDG-SW3 cells were subjected to FISH with a probe for lncRNA953Rik (red) (**bottom panel**) or without any probe (**top panel**). DAPI staining was used to visualize nuclei (blue). Scale bar: 5 μm. (**B**) In vitro-transcribed lncRNA953Rik or a negative control RNA of the same length was incubated with whole-cell lysates from IDG-SW3 cells. After pull-down of each RNA, RNA-binding proteins were identified through LC-MS/MS analysis. The Venn diagram shows the numbers of the proteins identified. The top five proteins that specifically bound to lncRNA953Rik are listed. (**C**) RNA pull-down samples obtained as described in (**B**) were used for immunoblotting to confirm the binding of lncRNA953Rik to CCAR2. The arrow shows the location of CCAR2. (**D**) In vitro-transcribed lncRNA953Rik or the negative control RNA of the same length was incubated with nuclear extracts from IDG-SW3 cells. After pull-down of each RNA, the samples were used for immunoblotting to confirm the binding of lncRNA953Rik to CCAR2 in nuclei. The arrow shows the location of CCAR2. (**E**) Nuclear extracts from IDG-SW3 cells were subjected to immunoprecipitation with an anti-CCAR2 antibody or control IgG, followed by qPCR with primers targeting lncRNA953Rik (**left**). Error bars, SDs; *n* = 4; *t* test; **, *p* < 0.01 versus control. Agarose gel electrophoresis of the PCR products (**right**). Values under each group of immunoblot analysis are relative protein levels measured by densitometry.

**Figure 4 ijms-24-13633-f004:**
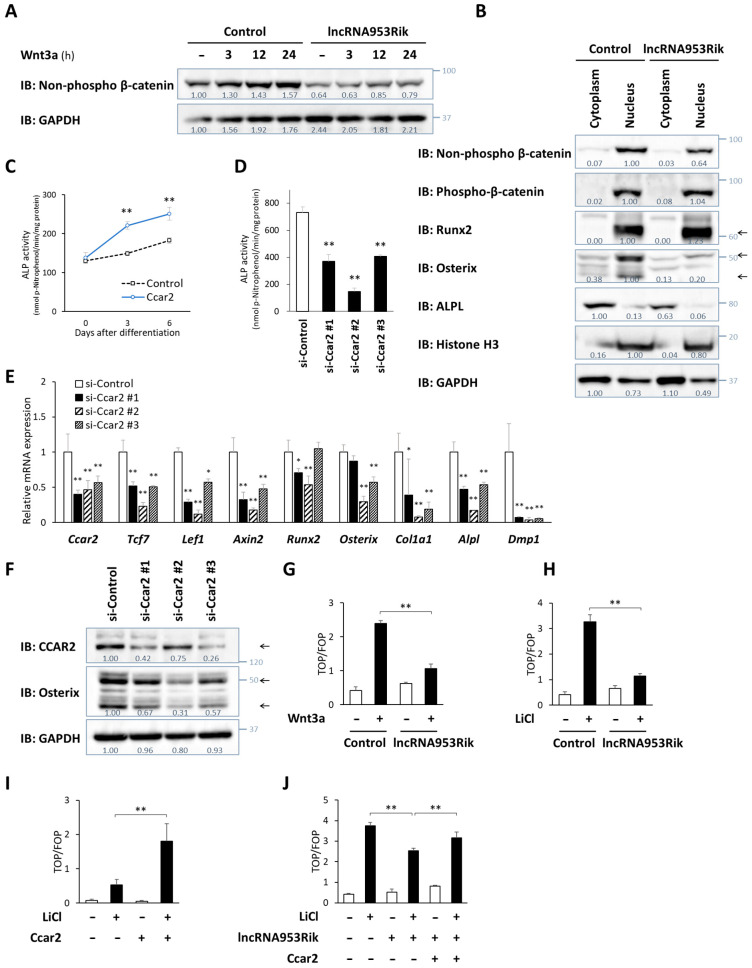
lncRNA953Rik suppresses Wnt/β-catenin signaling. (**A**) Immunoblot analysis of non-phospho-β-catenin in whole-cell lysates from lncRNA953Rik-overexpressing or mock IDG-SW3 cells (control) exposed to 100 ng/mL Wnt3a for the indicated times (up to 24 h). (**B**) Immunoblot analysis of non-phospho/phosphor-β-catenin and marker genes of bone formation in cytoplasmic and nuclear extracts from lncRNA953Rik-overexpressing or mock IDG-SW3 cells (control). The arrows show the locations of Runx2 and Osterix. (**C**) The *Ccar2* plasmid or empty plasmid was transfected into IDG-SW3 cells. ALP activity was measured over 6 days of culture with differentiation medium. Error bars, SDs; *n* = 3; *t* test; **, *p* < 0.01 versus control. (**D**–**F**) Differentiation of IDG-SW3 cells was induced on day 0. An siRNA targeting *Ccar2* (#1, #2 or #3) or a negative control siRNA was transfected on days 0, 3 and 6. Samples were obtained on day 9 and subjected to an ALP activity assay (**D**), qPCR (**E**) and immunoblot analysis (**F**). Error bars, SDs; *n* = 5; ANOVA followed by Dunnett’s test; **, *p* < 0.01 versus control (**D**). The indicated RNA expression levels were normalized to the level of *Gapdh*. Error bars, SDs; *n* = 3; ANOVA followed by Dunnett’s test; *, *p* < 0.05; **, *p* < 0.01 versus control (**E**). The arrows show the locations of CCAR2 and Osterix (**F**). (**G**,**H**) The TOP-Flash or FOP-Flash reporter plasmid and the Renilla luciferase plasmid were transfected into lncRNA953Rik-overexpressing or mock IDG-SW3 cells (control) on day 0. The cells were cultured with or without 100 ng/mL Wnt3a (**G**) or 20 mM LiCl (**H**) from day 1 to day 4. Samples were obtained on day 4 and subjected to a luciferase reporter assay. *n* = 3. (**I**) The *Ccar2* plasmid or empty plasmid, the TOP-Flash or FOP-Flash reporter plasmid, and the Renilla luciferase plasmid were transfected into IDG-SW3 cells on day 0. The cells were cultured with or without 20 mM LiCl from day 1 to day 2. Samples were obtained on day 2 and subjected to a luciferase reporter assay. *n* = 4. (**J**) lncRNA953Rik, the *Ccar2* plasmid or empty plasmid, the TOP-Flash or FOP-Flash reporter plasmid, and the Renilla luciferase plasmid were transfected into IDG-SW3 cells on day 0. The cells were cultured with or without 20 mM LiCl from day 1 to day 5. Samples were obtained on day 5 and subjected to a luciferase reporter assay. *n* = 6. (**G**–**J**) Transcriptional activity is shown as a ratio of the TOP-Flash to the FOP-Flash signal; each signal was normalized to the Renilla luciferase signal. Error bars, SDs; ANOVA followed by the Student-Newman-Keuls (SNK) test; **, *p* < 0.01. Values under each group of immunoblot analysis are relative protein levels measured by densitometry.

**Figure 5 ijms-24-13633-f005:**
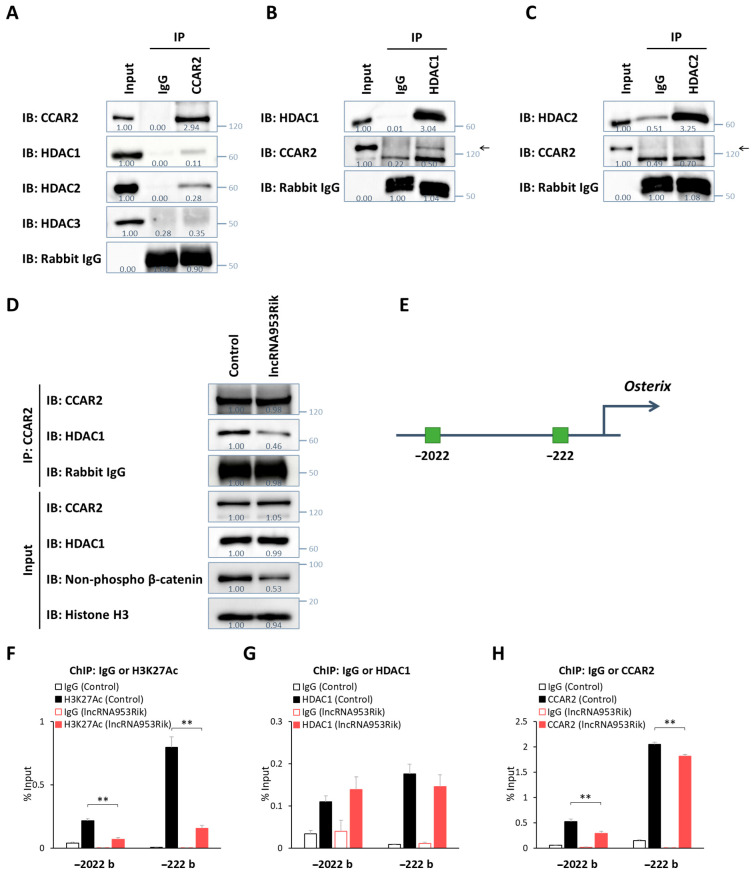
lncRNA953Rik suppresses *Osterix* transcription by sequestering CCAR2 from HDAC1. (**A**) Nuclear extracts from IDG-SW3 cells were subjected to immunoprecipitation with an anti-CCAR2 antibody or control IgG, followed by immunoblotting with an anti-HDAC1, anti-HDAC2 or anti-HDAC3 antibody. (**B**) Nuclear extracts from IDG-SW3 cells were subjected to immunoprecipitation with an anti-HDAC1 antibody or control IgG, followed by immunoblotting with an anti-CCAR2 antibody. The arrow shows the location of CCAR2. (**C**) Nuclear extracts from IDG-SW3 cells were subjected to immunoprecipitation with an anti-HDAC2 antibody or control IgG, followed by immunoblotting with an anti-CCAR2 antibody. The arrow shows the location of CCAR2. (**D**) Nuclear extracts from lncRNA953Rik-overexpressing or mock IDG-SW3 cells (control) were subjected to immunoprecipitation with an anti-CCAR2 antibody, followed by immunoblotting with an anti-HDAC1 antibody. (**E**) Schematic diagram of TCF/LEF-binding sites in the *Osterix* promoter region. (**F**–**H**) lncRNA953Rik-overexpressing and mock IDG-SW3 cells (control) were subjected to ChIP with an anti-H3K27ac, anti-HDAC1 or anti-CCAR2 antibody, followed by qPCR using primers targeting the *Osterix* promoter region shown in (**E**). Error bars, SDs; *n* = 4; ANOVA followed by the SNK test; **, *p* < 0.01. Values under each group of immunoblot analysis are relative protein levels measured by densitometry.

**Figure 6 ijms-24-13633-f006:**
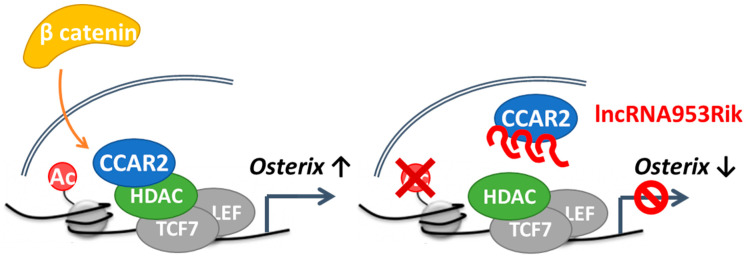
Model showing the mechanism by which lncRNA953Rik inhibits bone formation. When β-catenin is translocated into the nucleus, CCAR2 inhibits HDAC1, resulting in maintenance of H3K27 acetylation in the *Osterix* promoter region, in turn leading to transcriptional upregulation of *Osterix* (**left**). When lncRNA953Rik is abundant, it binds to CCAR2 and sequesters CCAR2 from HDAC1, resulting in uninhibited HDAC1 activity, leading to deacetylation of H3K27 in the *Osterix* promoter region and the consequent transcriptional downregulation of *Osterix* (**right**).

**Table 1 ijms-24-13633-t001:** The sequences of the primers for qPCR.

Gene	Forward/Reverse	Sequence
*2310069G16Rik*	Forward	5′-AATAATCACGTGGTGCGGCAG-3′
	Reverse	5′-CCGCCGCACGTGTTCCGAAGCCC-3′
*lncRNA953Rik*	Forward	5′-CATGCGAGGGACTGCTGAT-3′
	Reverse	5′-GGTGCTGAGAAGGCAAAGAT-3′
*A330033J07Rik*	Forward	5′-GTTGAGCGCGATAATGCAAA-3′
	Reverse	5′-CTCCATAAGCTGTGCGTTGA-3′
*Alpl*	Forward	5′-TTCCTGGGAGATGGTATG-3′
	Reverse	5′-TTATATGTCTTGGAGAGGGC-3′
*Axin2*	Forward	5′-TGTGAGATCCACGGAAACAG-3′
	Reverse	5′-TGGCTGGTGCAAAGACATAG-3′
*Ccar2*	Forward	5′-GAGGATCAACCCACTTCC-3′
	Reverse	5′-AAGACTCGCTGCTTTTCC-3′
*Col1a1*	Forward	5′-AGATGTAGGAGTCGAGGGAC-3′
	Reverse	5′-CATAGCCATAGGACATCTGG-3′
*Dkk1*	Forward	5′-TTGACAACTACCAGCCCTAC-3′
	Reverse	5′-GAAAATGGCTGTGGTCAG-3′
*Dmp1*	Forward	5′-AAGAGAGGACGGGTGATTTG-3′
	Reverse	5′-TCCGTGTGGTCACTATTTGC-3′
*Gapdh*	Forward	5′-ACCCAGAAGACTGTGGATGG-3′
	Reverse	5′-ACCCAGAAGACTGTGGATGG-3′
*Gm6634*	Forward	5′-CTTGGTTAAAGTGGATTACATTGAT-3′
	Reverse	5′-TGCTGAAGATGTCACCACTG-3′
*H19*	Forward	5′-GCCTTCTTGAACACCATGG-3′
	Reverse	5′-CAGACATGAGCTGGGTAGCA-3′
*Krm1*	Forward	5′-ACCGAGTGCAATAGTGTC-3′
	Reverse	5′-GAGTCCCTGATATCAAACAG-3′
*Krm2*	Forward	5′-GCGCATAACTTCTGTAGG-3′
	Reverse	5′-CTTTCAGAGCCACAGAAG-3′
*Lef1*	Forward	5′-GAAGGAAAGCATCCAGAC-3′
	Reverse	5′-GGCACTTTATTTGATGTCC-3′
*Meg3*	Forward	5′-CCTGGATTAGGCCAAAGCC-3′
	Reverse	5′-AGTCTTGGGTCCAGCATGTC-3′
*Mirg*	Forward	5′-CCTCTGCTGGACAGCTTCAG-3′
	Reverse	5′-CATAGGCAGGGTTCCTTGAA-3′
*Osteocalcin*	Forward	5′-TCTGACAAAGCCTTCATGTCCA-3′
	Reverse	5′-CGGTCTTCAAGCCATACTGGTC-3′
*Osterix*	Forward	5′-GTCCTCTCTGCTTGAGGA-3′
	Reverse	5′-AGGAGAGAGGAGTCCATTG-3′
*Runx2*	Forward	5′-GCCGGGAATGATGAGAACTA-3′
	Reverse	5′-ATGCGCCCTAAATCACTGAG-3′
*Sfrp1*	Forward	5′-TGCTCAAATGTGACAAGTTC-3′
	Reverse	5′-CAGCTTCAAGGGTTTCTTC-3′
*Sfrp4*	Forward	5′-TATGATGGTGCAAGAAAG-3′
	Reverse	5′-TAGGTGACAAAGACTTGAAG-3′
*Tcf7*	Forward	5′-AGCTTTCTCCACTCTACG-3′
	Reverse	5′-GAGGTCAGAGAATAAAATCC-3′
*Wif1*	Forward	5′-TGGTCTGTGTGTCACTCC-3′
	Reverse	5′-GCATTTACCTCCATTTCG-3′

**Table 2 ijms-24-13633-t002:** The sequences of the antisense oligonucleotides.

Antisense Oligonucleotide	Sequence
Anti-lncRNA953Rik #1	5′-CTTGCGTCTTAAATTC-3′
Anti-lncRNA953Rik #2	5′-TCATGCGTTAACTTGC-3′
Anti-lncRNA953Rik #3	5′-ACAGGTCATTAAGGAC-3′
Negative control	5′-AACACGTCTATACGC-3′

**Table 3 ijms-24-13633-t003:** The sequences of the siRNAs.

siRNA	Forward/Reverse	Sequence
si-Ccar2 #1	Forward	5′-AGAGGCUACUGAACAGGCUCCUGAU-3′
	Reverse	5′-AUCAGGAGCCUGUUCAGUAGCCUCU-3′
si-Ccar2 #2	Forward	5′-CCUCAUCAAUGUGGGAAGCCUGUUA-3′
	Reverse	5′-UAACAGGCUUCCCACAUUGAUGAGG-3′
si-Ccar2 #3	Forward	5′-GCGGGAUGAUGGAGAGGACGAAUUU-3′
	Reverse	5′-AAAUUCGUCCUCUCCAUCAUCCCGC-3′

**Table 4 ijms-24-13633-t004:** The sequences of the primers.

Gene	Forward/Reverse	Sequence
*lncRNA953Rik*	Forward	5′-GATCACTAATACGACTCACTATAGGGAATCTCAAGGAAGCTCTTT-3′
	Reverse	5′-TTCAATTACCATGGAGGTTTA-3′
*Negative control*	Forward	5′-GATCACTAATACGACTCACTATAGGGAACTACGGCTACACTAGAA-3′
	Reverse	5′-TTACTTTTAAAGTTCTGCTATGT-3′

**Table 5 ijms-24-13633-t005:** The sequences of the primers.

Location	Forward/Reverse	Sequence
*Osterix* −222 bases	Forward	5′-CTCATTGGATCCGGAGTCTTCT-3′
	Reverse	5′-TGTCTGTAGGGATCCACCCTCTA-3′
*Osterix* −2022 bases	Forward	5′-TAGAGCATTCCCTTTGGG-3′
	Reverse	5′-AGCAGGACTGAGGGACAG-3′

## Data Availability

Data will be made available upon request.

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
