# Peer review of "A Novel Long Noncoding RNA in Osteocytes Regulates Bone Formation through the Wnt/β-Catenin Signaling Pathway"

_ijms, 2023, doi:10.3390/ijms241713633_

Round 1

Reviewer 1 Report

The manuscript by Makoto Arai et al entitled “A Novel Long Noncoding RNA in Osteocytes Regulates Bone Formation through the Wnt/β-Catenin Signaling Pathway is scientifically sound and well written. The study design is scientifically convincing and well-conducted. Introduction has adequate information about the previous study findings for readers to follow the present study rationale. However, I have few concerns

1.     Authors should disclose criteria for selecting lncRNA953Rik among the group of lncRNAs whose overexpression leads to a significant decrease in mineralization (Figure 1D)

2.     Densitometry of all blots should be performed

3.     Authors should provide details about any bioinformatic tool that is being used to predict putative binding sites of a lncRNA to a protein (if any)

4.     Authors disclosed that lncRNA953Rik sequestered CCAR2 from HDAC1, leading to deacetylation of H3K27 in the Osterix promoter and consequent transcriptional downregulation of Osterix. Is there any microRNA that is being sponged by lncRNA953Rik that plays a role in regulation of CCAR-HDAC1 axis

Needs minor editing

Reviewer 2 Report

The authors Arai et al. identified the role of a lncRNA in osteocytes, and their findings can pave the way for novel therapeutic options targeting lncRNAs in osteocytes to treat bone metabolic diseases such as osteoporosis. Overall, the ms is quite interesting and well supported by their results and explanation. I have no such major concern about this ms, but till some minor issues need to address to improve the ms before acceptance.

Minor comment:

Author need to address some background about lncRNA for better understanding of reader.

Reviewer 3 Report

Referee’s comment

Article n° IJMS 258344

Title: A Novel Long Noncoding RNA in Osteocytes Regulates Bone  Formation through the Wnt/β-Catenin Signaling Pathway

Authors:   Arai M,  Ochi H,  Sunamura S,  Ito N,  Nangaku M, Takeda S,  Sato S.

General comment

The study by  Arai et al. aims to elucidate the role of  lncRNA953Rik, a long noncoding RNA in the regulation of of bone formation. By the obtained data they found  that  lncRNA953Rik, by acting through  Wnt/β-28 catenin signaling was able to suppress osteogenic differentiation. Authors concluded that lncRNA953Rik could be exploited as a therapeutic  target in the Wnt/β-catenin signalling pathway and bone formation. Therefore this lncRNA could be usefull for the development of novel therapeutic options targeting osteocytes for treating diseases such as osteoporosis. The subject is of interest and the experiments appear to have been carried out quite well. The results also appear to be new and promising. However, I have a few questions and suggestions.

Specific points

1)      To give greater prominence and value to the manuscript I think it is useful to show a dose response with  lncRNA953Rik. At least three doses it is necessary to use, one dose is insufficient to obtain a dose-dependent effect.

2)      The authors report that the targets of lncRNA953Rik are about 245 and among these CCAR2 is in 2nd place. In the first one, from the diagram there appears to be PAR14. Why was attention focused on CCAR2 and not on PAR14? What is the rationale for this choice?

3)      The osterix gene can be activated by many mediators, such as MAPK, ERK, RUNX2 and can be inhibited by many miRNAs (see Front. Cell Dev. Biol., 15 December 2020 Sec. Molecular and Cellular Pathology Volume 8 - 2020 https://doi.org/10.3389/fcell.2020.601224). How do you explain your results in the light of all these actions by these molecules always present in the extracellular matrix?

4)      Materials and methods section: paragraph 4.2; reference n.46 is useless since it makes no reference to the ALP method. If this is a way of self-quoting it is not exercised in the right way.

5)      Materials and methods section: paragraph 4.4; reference n.47 is useless since it makes no reference to the Alizarin red S staining  method. Same reasoning as point 4.

6)      Materials and methods section: paragraph 4.18; references n.51-54 are useless since they make no reference to the Luciferase assay  method. On the other hand, the method used is based on a commercial kit that does not require references.

There are some errors
